# Treatment of a Large Defect Induced by Atrophic Nonunion of Femoral Fracture in a Dog with Autogenous Coccygeal Bone Grafting

**DOI:** 10.3390/vetsci10060388

**Published:** 2023-06-07

**Authors:** Kyuman Cho, Kilsang Lee, Kyungsik Kang, Minkyung Kim

**Affiliations:** 1Chokyuman Vet Surgical Center, Bucheon 14624, Republic of Korea; kyuman@hanmail.net (K.C.);; 2Veterinary Medical Research Institute, Jeju National University, Jeju 63243, Republic of Korea; 3Keunmaum Animal Medical Center, Busan 48096, Republic of Korea; 4Department of Veterinary Surgery, College of Veterinary Medicine, Gyeongsang National University, Jinju 52828, Republic of Korea

**Keywords:** nonunion, femoral fracture, autograft, coccygeal bone, platelet-rich plasma, dog

## Abstract

**Simple Summary:**

This study addressed the complexities of nonunion in small animal orthopedics and highlighted the potential of autogenous corticocancellous bone grafts to facilitate bone healing. Autogenous cancellous bone grafting has proven to be the most effective means of stimulating fracture healing in orthopedics; however, filling large bone defects with cancellous bone can be challenging. In these cases, cortical bone grafts may provide mechanical stability to the defective area. This case study investigated the use of the coccygeal bone as an autogenous bone graft to fill a large femoral defect. To prevent fixation failure, rigid fixation was achieved between the coccygeal and femoral bone fragments using orthogonal locking plates and screws, strengthening the mechanical structure. Harvesting was performed immediately before transplantation to increase the viability of the grafted bone. Various methods can be employed to enhance osteoinductivity and osteoconductivity to promote successful engraftment, including the use of platelet-rich plasma and active rehabilitation to improve the blood supply and increase biological activity. Although coccygeal bone grafting has been reported in the radius and tibia, this is the first instance of the coccygeal bone being used as an autogenous bone graft in femoral fracture nonunion.

**Abstract:**

An 11-month-old castrated male Pomeranian presented with nonunion following a femoral fracture that occurred after femoral head and neck osteotomy. Radiography and computed tomography revealed severe atrophy of the proximal bone fragment and retardation of the ipsilateral distal fragment and tibia. An autogenous bone graft using coccygeal bone was performed, in which three and a half coccyges were placed in succession and fixed using an orthogonal locking plate. To promote bone healing and facilitate proper weight bearing and ambulation, bone morphogenetic proteins, biphasic calcium phosphate, platelet-rich plasma, passive-range-of-motion exercises, transcutaneous electrical nerve stimulation, neuromuscular electrical stimulation, and low-level laser therapy were applied. During the four-year follow-up, it was observed that the previously engrafted bone healed well and maintained stability over a prolonged period, resulting in the patient being able to walk comfortably with good outcomes. However, some degree of lameness was noted in the dog when running owing to limb shortening and joint contracture.

## 1. Introduction

The treatment of nonunions is one of the most challenging areas in small animal orthopedics. The diagnosis of fracture nonunion is made using radiographic evidence of bone hypertrophy or atrophy of bone fragments, defects between the fracture ends, sclerosis, and a closed medullary cavity [1]. Most nonunions are the result of inadequate decision making and technical mistakes by the surgeon, rather than factors associated with the patient or its owner [2]. In small-breed dogs, nonunion has the highest incidence in the radius, followed by the tibia, whereas the femur has the lowest occurrence, which could be attributed to the amount of soft tissue and blood supply [3]. An optimal bone graft not only supplies osteogenic cells for new bone formation, but also contains osteoinductive factors and an osteoconductive matrix that stimulates the differentiation of undifferentiated mesenchymal stem cells. The osteoconductive matrix also provides mechanical support for the internal growth and organization of the new bone structure [4]. Autogenous cancellous bone grafting is the most effective treatment and accepted as the gold standard for stimulating fracture healing in orthopedics [5]. Autogenous cancellous bone grafting is effective in most cases; however, if the defect is large owing to bone resorption or infection, it can be challenging to fill the cavity with cancellous bone, because cancellous bone grafts do not provide mechanical stability. In cases with large defects, cortical bone grafts provide mechanical stability to the defect area [6].

## 2. Case Presentation

### 2.1. History and Clinical Examination

A castrated 11-month-old Pomeranian male, weighing 3.29 kg, presented with non-weight-bearing lameness in the left hind limb. The dog underwent femoral head and neck osteotomy at another hospital when he was six months old due to a femoral head fracture. Approximately one month later, the dog fractured his proximal femur. The dog underwent fracture repair surgery with a plate and screws; however, the bone fracture failed to heal. Consequently, the patient underwent a second surgery, which again failed, resulting in severe atrophic nonunion. Therefore, the dog was referred for revision surgery.

During the physical examination, the range of motion of the stifle and tarsal joints was found to be severely decreased. Radiography revealed implant failure in the left femur, severe atrophy of the proximal bone fragment, and retardation of the ipsilateral distal bone fragment of the femur and tibia compared to the contralateral side (Figure 1A,B). The failed implants were removed and computed tomography (CT) was performed for preoperative surgical planning (Figure 1C). An autogenous corticocancellous bone graft was planned and the coccygeal bone was ultimately chosen as the autograft (Figure 2).

### 2.2. Anesthesia and Surgical Treatment

Before the surgical procedure, the patient received intravenous (IV) administration of premedicants and prophylactic antibiotics containing acepromazine (0.05 mg/kg, IV), ketamine (5 mg/kg, IV), and cefazolin (25 mg/kg, IV). Anesthesia was induced with propofol (6 mg/kg, IV) and maintained with isoflurane.

To harvest the planned 8th to 11th coccygeal bones, caudectomy was first performed, and the bone was separated from the surrounding soft tissues. The articular cartilage at both ends of the bone was resected using an oscillating saw. During the surgical procedure, the soft tissue contracture was severe and required stretching. A lateral incision was made to access the left femur, revealing a poorly vascularized proximal end of the distal bone fragment. To ensure proper healing and an adequate blood supply, the end of the bone was decorticated, rimmed, and trimmed. The proximal fragment remained in situ, adjacent to the coccygeal bones. The medial surface of the proximal bone was carefully debrided with an orthopedic rasp to promote the healing of the coccygeal bone. To apply the graft, three and a half coccygeal bones were prepared in succession and fixed with a 1.2 polyaxial locking plate system (ARIX^®^, Jeil Medical Co., Seoul, Republic of Korea) for mechanical stability (Figure 3A). Orthogonal plating was performed laterally and cranially to ensure the stability of the proximal, distal, and coccygeal bones (Figure 3B). Each coccygeal bone was stabilized with two or three screws, and screws inserted through the lateral side fixed both the proximal and coccygeal bones together. After thoroughly washing the area, recombinant human bone morphogenetic protein-2 (BMP) and biphasic calcium phosphate (BCP) (Cowell BMP^®^, Cowellmedi, Busan, Republic of Korea) were applied around the gap between the coccygeal bones and the coccygeal and distal bone fragments to encourage healing (Figure 3C). The surgical site was closed in a standard manner without using a bandage to promote movement.

### 2.3. Postoperative Management and Prognosis

During anesthesia, 10 mL of blood with citrate was collected from the patient to prepare platelet-rich plasma (PRP) (Sell Neo PRP 10cc kit^®^, NeoGenesis Co., Ltd., Seoul, Republic of Korea) for application to the soft tissue around the fracture sites to enhance the healing environment. PRP was administered three times with a one-week interval between each application, resulting in the administration of 1.5 mL of PRP each time. Transcutaneous electrical nerve stimulation (TENS) and neuromuscular electrical stimulation (NMES) were performed twice per week, and low-level laser therapy (LLLT) was performed daily until hospital discharge. Additionally, passive-range-of-motion (PROM) exercises were conducted on the hip, stifle, and tarsal joints four times a day until the patient was able to bear weight. The patient received postoperative oral medication administered twice daily, including cephalexin (25 mg/kg), famotidine (0.5 mg/kg), and carprofen (2.2 mg/kg). The patient was discharged on day 40 following the bone grafting surgery.

Immediately after surgery, the patient exhibited knuckling and was unable to bear weight. However, a neurological examination revealed no nerve damage. Limb shortening compared with the contralateral side was observed after the operation. Although the patient could partially bear weight, knuckling persisted for six weeks after the operation. Three months postoperatively, the patient was able to fully bear weight in the limb-extended position, with complete resolution of knuckling. On radiographic follow-up, the grafted bone showed temporarily increased lucency four weeks postoperatively, which was resolved three months after the operation (Figure 4B,C). Eighteen months postoperatively, radiographic examination revealed complete bone healing; however, loosening of the proximal second screw in the lateral plate was observed (Figure 4D). Although the owner was advised to undertake removal of the plate and screw immediately after confirming proximal screw loosening and bone healing completion, they refused. The patient was followed up for four years after the operation and exhibited a satisfactory gait during walking and trotting; however, some lameness was noted following intense exercise or galloping.

## 3. Discussion

The patient in this case had hip contractures and muscle atrophy from two previously failed surgeries, which made the biological odds unfavorable. Autogenous bone grafting was performed to aid bone healing. While there are no large studies on the complications following autologous bone harvesting in veterinary medicine, complications as high as 25% have been reported in humans, including pain, sepsis, fatigue fractures, intraoperative bleeding, prolonged surgery, and insufficient bone harvesting [7]. In dogs, the iliac crest, proximal humerus, and proximal tibia are considered the most abundant sources of autogenous cancellous bone [8]. Owing to the substantial size of the defect, we needed to determine the donor site that would allow us to harvest the largest amount of bone with the fewest complications, while still being able to fill the large defect with autologous bone. Hence, the coccygeal bone was selected as the autogenous bone graft. Furthermore, due to the differences in the coccygeal bone in dogs from that in humans, dogs with short or almost no tails due to breed characteristics, tumors, or fractures can live without major inconvenience.

The successful use of the coccygeal bone to fill large defects, including the tibia and radius, has also been reported. One case report demonstrated a radial fracture nonunion, in which two coccygeal bones similar in diameter to the radius were selected and fixed in succession [9]. In our case, we also arranged the bones in succession, as described by Choi and Yoon (2022); however, unlike in the previous study, we inserted screws into the grafted bone itself to increase fixation because there was no place for the plate to be fixed at the proximal fragment, and the length of the graft was too long. By applying the orthogonal double-locking plate to the patient, we could prevent the fixation from collapsing and strengthen the mechanical structure. Another case report presented a tibial fracture nonunion, in which the coccygeal bone was placed parallel to fill the defect owing to the difference in thickness between the tibia and coccygeal bone [10]. In the present case, we grafted the coccygeal bone onto the femur. Although the thickness and diameter of the coccygeal bone are usually much smaller than those of the femur, in our case, the diameter of the femur was significantly reduced owing to bone atrophy and retardation, resulting in a small difference in diameter between the coccygeal bone and femur.

Cancellous bone grafts have a viability rate of 85–100%, which decreases over time. Therefore, the graft bone should be harvested immediately before transplantation to ensure maximal graft viability. In experiments with rabbit cancellous bone, viability decreased to 57% 3 h after harvesting and was further reduced when preserved in cooled saline [11]. In our case, we performed a caudectomy immediately before femoral fracture surgery to improve the viability of the grafted coccygeal bone. Two operators performed the surgery simultaneously to expedite transplantation, with one approaching the femoral surgical site and the other separating the coccygeal bone from the surrounding soft tissues. By minimizing the time between bone harvesting and grafting, we may have ensured the survival of the grafted bone.

In some cases, autogenous bone grafting procedures may fail to achieve proper healing because of sequestration or resorption of the grafted bone [2]. To address this issue, various methods have been proposed to promote successful engraftment, including the use of BMP, BCP, and PRP [2,12,13,14,15]. BMP enhances the osteoinductivity of autologous bone grafts and stimulates osteoblast proliferation, whereas BCP exhibits osteoconductivity and promotes the proliferation of osteogenic cells [2,14]. Combining BMP with autologous bone grafting can improve bone healing more effectively than using them separately [14]. Furthermore, clinical studies have demonstrated that PRP can enhance the blood supply and increase biological activity for accelerated bone recovery [13]. To increase the engraftment success rate, we applied BMP and BCP along with autologous coccygeal bone and injected PRP into the surgical site. In addition, we also administered rehabilitation treatments, such as TENS, NMES, LLLT, and PROM, to improve blood supply and enhance biological activity, thereby promoting bone healing.

In the early stages of direct bone healing, also known as primary bone healing, radiographic fracture lines remain visible. During contact healing, there was no resorption of the fragment tip, and the cutting cone progressed across the fracture gap and became radiographically opaque. In clinical cases of simple transverse fractures treated with rigid fixation with compression, the fracture line may slowly disappear without forming callus [16,17]. In our case, as shown in Figure 3B, partial absorption of the grafted bone was observed 4 weeks after surgery, and the osteotomy line between the grafted coccygeal bone and the distal bone fragment was visible radiographically. However, in Figure 3C, which was taken 3 months later, the previously seen lucency is opaque, and in Figure 3D, which was taken after 8 months, the osteotomy line has completely disappeared. Despite the extended time required for complete bone healing, rigid fixation was maintained for an extended period using locking plates and screws, which facilitated healing without callus formation.

According to D. Franczuszki [18], a >20% difference in femur length can lead to gait problems. Following the operation, the patient exhibited a 22.8% reduction in femur length, a 12.0% decrease in tibia length, and a 40.0% reduction in tibia diameter compared with the normal contralateral limb. This was because of a fracture and nonunion that occurred during the juvenile period. The patient was able to walk, but due to the overall shortening of the limb, the limb was extended, and although there was significant improvement with rehabilitation, the range of motion remained limited due to the presence of joint contracture.

Significant muscle contracture from prolonged nonunion made surgery challenging. We measured the minimum length needed for the bone graft preoperatively, aiming for a difference within 20% compared to the contralateral limb. During surgery, the muscles were stretched by gripping and carefully pulling the bone fragments in opposite directions for several minutes using serrated bone-holding forceps. The sciatic nerve and muscle condition were monitored throughout to prevent damage. The intraoperative muscle stretching was beneficial; however, achieving the originally planned bone graft length was impossible due to severe muscle contracture. We could only accommodate three and a half coccygeal bones, resulting in a limb length difference exceeding 20%.

## 4. Conclusions

Despite some limitations, the use of the coccygeal bone as an autogenous bone graft material has proven to be an excellent option for treating femoral fracture nonunion. In addition to proper mechanical fixation, various supplementary methods, such as BMP, BCP, PRP, PROM, TENS, NMES, and LLLT, can be useful in enhancing the engraftment rate of autologous bone. This case represents the first report of coccygeal bone grafting into a femoral nonunion; the grafted bone healed successfully, eventually enabling the patient to regain mobility.

## Figures and Tables

**Figure 1 vetsci-10-00388-f001:**
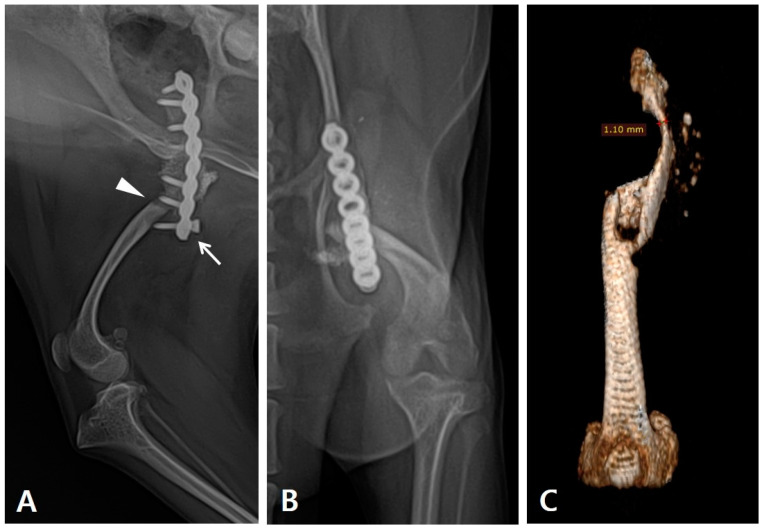
Preoperative evaluation and planning carried out using radiography and CT scans. (**A**) Lateral and (**B**) craniocaudal radiograph of the patient’s left femur. Re-fracture occurred (arrowhead) with the distal-most screw pulled out (arrow). Underdeveloped femur and tibia were observed. (**C**) The three-dimensional reconstructed CT image of the left femur after implant removal shows severe atrophy in the proximal bone fragment, with a 1.10 mm thickness at its thinnest point.

**Figure 2 vetsci-10-00388-f002:**
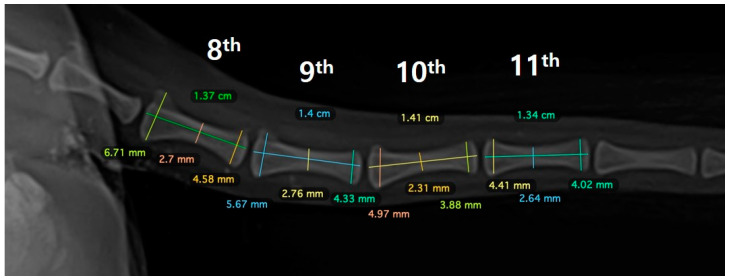
The preoperative surgical plan utilizing the 8th–11th bones of the coccyx.

**Figure 3 vetsci-10-00388-f003:**
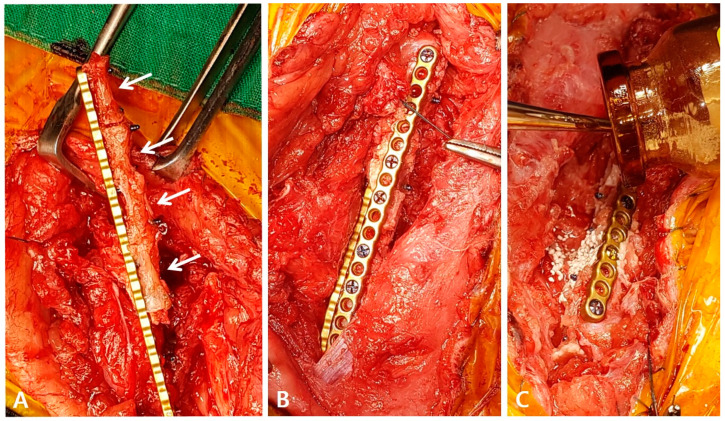
Intraoperative coccygeal bone grafting with locking plates. (**A**) Harvested coccygeal bones were trimmed and placed in a row with minimal spacing, and then fixed with a cranial polyaxial locking plate (arrows). (**B**) After securing the coccygeal bone and its distal fragment with a cranial-side plate, the proximal bone fragment was first fixed laterally to the coccygeal bone, and then fixed to the distal fragment. (**C**) BMP and BCP were applied around the gap between the coccygeal bones and the coccygeal and distal bone fragments.

**Figure 4 vetsci-10-00388-f004:**
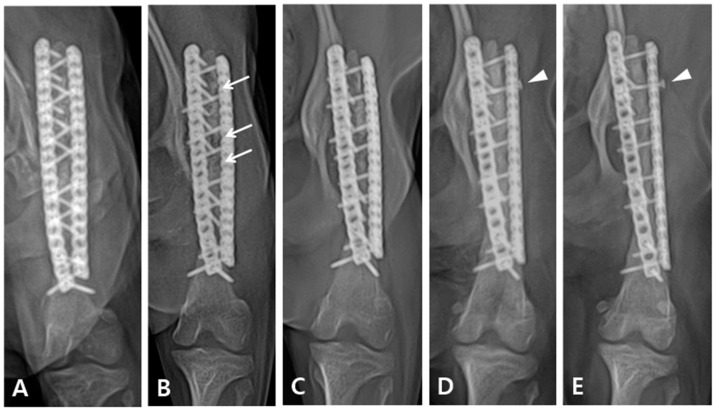
Follow-up radiography after the operation. (**A**) Two weeks after the operation. (**B**) Four weeks after the operation. Some bone resorption observed in the grafted area (arrow). (**C**) Three months after the operation, the parts that were radiolucent at the last follow-up appeared more opaque. (**D**) Eighteen months after the operation, the cortex exhibited continuity and there was no visible fracture line, indicating that the bones had completely fused. However, it was confirmed that a screw in the lateral plate had loosened (arrow head). (**E**) Four years after the operation, the results from the last follow-up were maintained.

## Data Availability

The data presented in the study are available on request from the corresponding author. The data are not publicly available due to ethical and privacy concerns.

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
