# Peer review of "Treatment of a Large Defect Induced by Atrophic Nonunion of Femoral Fracture in a Dog with Autogenous Coccygeal Bone Grafting"

_vetsci, 2023, doi:10.3390/vetsci10060388_

Round 1
Reviewer 1 Report
To date, regenerative surgery has made great strides, with a notable regard for the proteins that are inserted into the fracture.
This article well demonstrates the result of cutting-edge surgery with the help of specific conditions leading to a very interesting goal.
The manuscript has remarkable scientific form and proper surgical training.
Reviewer 2 Report
The clinical case is very challenging and complex and the surgical treatment option was relatively original, and it was successful.
It is well documented in terms of pre-, intra- and postoperative images, namely imaging and surgical technique. It is also based on a correct, complete and current bibliographic review on the subject.
However, an English revision will be required. It is also noted that the authors use terms that are more appropriate to Equine Medicine and Surgery and not so much to Companion Animals.
Correction suggestions:
Lines 37 and 38:
… being able to walk and trot comfortably with good outcomes. However, some degree of lameness 37 was noted during galloping owing to limb shortening and joint contracture - …being able to walk comfortably with good outcomes. However, some degree of lameness was noted when running owing to limb shortening and joint contracture.
Lines 50 and 51:
… but also contains osteoinductive cells… - … but also contains osteoinductive factors…
Lines 51 and 52:
…undifferentiated mesenchymal cells. - …undifferentiated mesenchymal stem cells.
Lines 69 and 124:
Substitution of “hock joints” by “tarsal joints” or “tarsus”
Reviewer 3 Report
This case report describes the attempt to facilitate bone healing in a dog with severe atrophy of the proximal bone fragment using autogenous corticocancellous bone grafts. The authors have gone to great lengths to achieve their goal, however, I have some concerns about their approach, such as the number of coccygeal bones that they used, the stretching of the muscles involved, the post-operative care, etc. Furthermore, the surgical procedure should be described more clearly. These and some other comments are listed below.
Line 62: Add the weight of the animal
Line 72: (Figure A) Maybe correct it to (Figure A, B)
Line 74: (Figure B) Correct it to (Figure C)
Line 92: During surgery, the authors found severe soft tissue contracture that required stretching. Which muscles were stretched and by what method? What about the quadriceps? Is this contracture the reason that they were not able to achieve a proper length of the reconstructed femur?
Line 95: No information is given about the proximal bone fragment. Was this part of the femur removed, left in situ in proximity to the coccygeal bones, or else?
Line 97: Why did you use three and a half coccygeal bones and not four or five, so the length of the femur would be adequate and not 20% less? Maybe in that way, the limping would be less profound. Why you did not calculate the proper length of the femur based on X-rays of the other one?
Line 119: You don’t mention anything about the post-operative treatment as far as antibiotics and analgesia are concerned. Please add relative information.
Line 123: How long was the animal hospitalized?
Line 127: Why did you use three and a half coccygeal bones and not four or five, so the length of the femur would be adequate?
Lines 142-145: In Figures D and E it seems that callus formation was not even along the reconstructed area. Is this the reason that the plates and screws were not removed?
Line 209: You should add some comments on why the locking plates and screws were not removed after complete bone healing
Line 212: Would not that be prevented if you used more coccygeal bones?
Line 214: Where is Figure 5?
Minor editing is needed
Round 2
Reviewer 3 Report
The manuscript has been improved. However, the significance of content is average, due to the limitations of the surgical procedure to reconstruct a femur of proper length.
Some minor editing might be needed.